# CONCEPT-SAE: ACTIVE CAUSAL PROBING OF VISUAL MODEL BEHAVIOR

## ABSTRACT

Standard Sparse Autoencoders (SAEs) excel at discovering a dictionary of a model's learned features, offering a powerful observational lens. However, the ambiguous and ungrounded nature of these features makes them unreliable instruments for the **active, causal probing** of model behavior. To solve this, we introduce **Concept-SAE**, a framework that forges semantically grounded *concept tokens* through a novel hybrid disentanglement strategy. We first quantitatively demonstrate that our dual-supervision approach produces tokens that are remarkably faithful and spatially localized, outperforming alternative methods in disentanglement. This validated fidelity enables two critical applications: (1) we probe the causal link between internal concepts and predictions via direct intervention, and (2) we probe the model's failure modes by systematically localizing adversarial vulnerabilities to specific layers. Concept-SAE provides a validated blueprint for moving beyond correlational interpretation to the mechanistic, **causal probing** of model behavior.

## 1 INTRODUCTION

The ultimate goal of mechanistic interpretability (Olah et al., 2018; 2020) is to reverse-engineer neural networks, moving from observing their behavior to understanding their internal algorithms. A dominant approach, exemplified by Sparse Autoencoders (SAEs) (Huben et al., 2023; Ramaswamy et al., 2023; Yeh et al., 2020), decomposes a model's internal activations into a dictionary of learned features. This technique has been widely applied in both vision (Zhang & Zhu, 2018; Stevens et al., 2025; Lim et al., 2024; Olson et al., 2025) and natural language (Shu et al., 2025; Huben et al., 2023), providing a powerful observational lens. However, this paradigm remains fundamentally correlational; it provides a list of a model's computational parts but lacks the instruments needed to actively probe how these parts causally interact to produce the model's behavior.

The primary barrier to a causal science of interpretability is the lack of semantically grounded and disentangled features. The dominant paradigm for analyzing SAEs requires subjective, post-hoc inspection of latent tokens and their corresponding activation maps (Gao et al., 2024; Paulo & Belrose, 2025; Marks et al., 2024; Härle et al., 2025). While one can perform interventions on these learned features, their ambiguous and potentially entangled nature makes it difficult to isolate specific causal factors. This ambiguity undermines the scientific rigor of resulting causal claims, preventing the formation of reliable, falsifiable hypotheses. Other attempts to enforce conceptual alignment also fall short. Concept Bottleneck Models (Koh et al., 2020), for instance, while powerful for final layer analysis, impose a restrictive low-dimensional bottleneck that is often unsuitable for preserving the rich, high-dimensional information present in intermediate layers. Meanwhile, concept embeddings (Espinosa Zarlenga et al., 2022; 2023), lacking direct supervision on their values and spatial locations, suffer from semantic drift and fail to disentangle from background features, once again precluding the clean interventions required for rigorous causal probing.

To bridge this gap, we introduce **Concept-SAE**, a framework that upgrades SAEs from passive dictionaries into instruments for **active causal probing**. Our core contribution is a hybrid disentanglement strategy that forges clean, semantically grounded handles necessary for reliable experimentation. We anchor **concept tokens** to human-defined concepts using a robust dual-supervision mechanism on both their existence and spatial localization, ensuring they are faithful and disentangled. Simultaneously, we retain unsupervised **free tokens** to capture residual information and

preserve the capacity for open-ended discovery. This design transforms the SAE into a validated instrument, enabling the direct, mechanistic probing of a model's causal structure. The main contributions of this work are as follows:

- We introduce **Concept-SAE**, a novel framework that fundamentally advances sparse autoencoders from passive observational tools into instruments for the **active causal probing** of visual model behavior.

- We provide extensive experimental validation showing that our dual-supervision strategy produces concept representations that are **remarkably faithful, spatially localized, and cleanly disentangled**, demonstrating superiority over alternative concept-based methods.

- We demonstrate that the high fidelity of our concept representations unlocks new capabilities for probing model behavior, including establishing causal links via **direct intervention** on predictions and **systematically localizing** adversarial vulnerabilities.

## 2 RELATED WORKS

**Model Interpretability with Sparse Autoencoders.** Sparse Autoencoders (SAEs) have emerged as powerful tools for mechanistic interpretability, building on the sparse coding hypothesis (Olshausen & Field, 1997) to address feature superposition. By training an SAE to sparsely reconstruct model activations, polysemantic signals can be disentangled into interpretable, monosemantic features (Sharkey et al., 2022; Huben et al., 2023). SAEs have been applied to MLPs and attention heads (Kissane et al., 2024) in visual (Gorton, 2024) and textual (Kantamneni et al., 2025; Mudide et al., 2025; Minegishi et al., 2025) models, with improvements in training stability and feature quality (Rajamanoharan et al., 2024). Extracted features now support tasks like discovering computational circuits (O'Neill & Bui, 2024) and model control for AI safety (Marks et al., 2023). However, previous SAE-based methods rely on passive, manual inspection of latent tokens, often yielding concepts that are sparse, unstable, and semantically entangled, with little control over which concepts are analyzed. In contrast, our approach actively test whether specific concepts are represented in the model. Our method explicitly disentangles concept tokens from free tokens through dual supervision and staged training, enabling precise and faithful concept representation while preserving the exploratory capacity of traditional SAEs.

**Incorporating Predefined Concepts into SAE.** Our approach extends sparse autoencoders by explicitly incorporating predefined concepts into their latent space. Since SAEs are trained to reconstruct internal features, constraining certain tokens to represent human concepts parallels prior efforts that inject concepts into the prediction process. A representative example is the Concept Bottleneck Model (CBM) (Koh et al., 2020; Rao et al., 2024), which supervises latent features to align with predefined concepts and forces predictions to pass through these human-understandable variables. Another line of work encodes concepts as latent embeddings (Espinosa Zarlenga et al., 2022; 2023). However, these approaches are primarily designed for the output layer. In intermediate feature spaces, binary bottlenecks struggle to disentangle fine-grained concepts and often cause semantic overlap (Espinosa Zarlenga et al., 2022), while embeddings easily drift from their intended semantics without direct supervision (Espinosa Zarlenga et al., 2022). To address these challenges, we supervise the values of concept tokens, anchoring them to visual evidence that captures both existence and spatial localization. A staged training strategy further enforces this separation, preventing leakage into free tokens while preserving their exploratory capacity.

## 3 METHODOLOGY

**Concept-SAE** endows SAE-based interpretability models with the capability of incorporating predefined concepts. The overall procedure begins with concept label generation, where ground-truth annotations are derived in the form of segmentation masks and existence scores for a predefined set of concepts. We then train a Concept Autoencoder, composed of a Concept Tokenizer and a Concept Aggregator: the tokenizer learns to extract concept representations from intermediate feature maps, while the aggregator reconstructs those features from the concept embeddings, thereby forming a fully invertible, concept-centric representation of the model's internal state. To capture residual information beyond the predefined concept space, we further introduce a Free Autoencoder, which

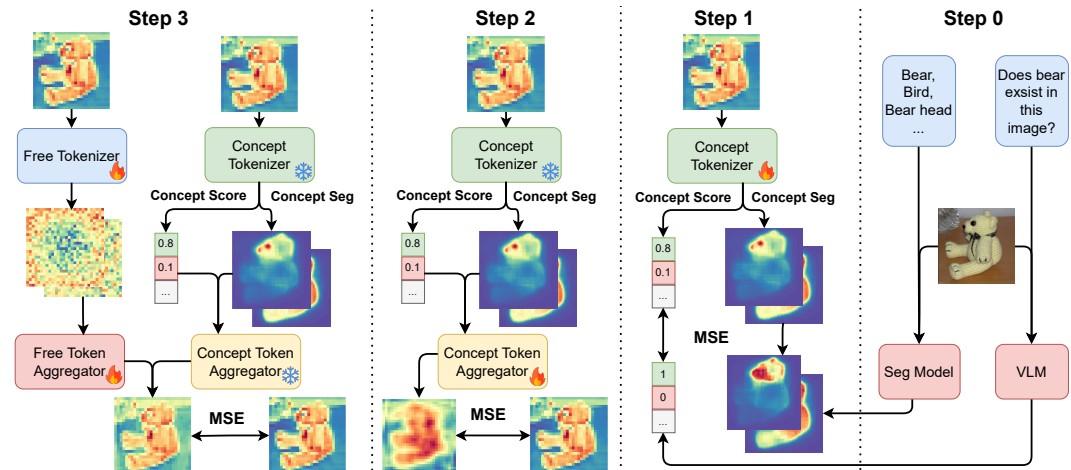

Figure 1: Overall training pipeline of our proposed method.

functions analogously to a conventional sparse autoencoder. The overall model training pipeline with four steps is shown in Fig. 1.

## 3.1 CONCEPT LABEL & SEGMENTATION GENERATION

To incorporate predefined concepts without constraining the SAE's ability to discover novel features, we adopt a minimal representation for each concept, consisting of its existence and spatial extent. This ensures that concept supervision anchors interpretation without injecting unnecessary information that could interfere with the SAE. We leverage two plug-and-play models to provide precise ground-truth annotations of these two signals: A vision-language model (VLM) determines whether each concept is present in the image, yielding a binary existence score. In parallel, a segmentation model produces an initial spatial mask. The two outputs are then fused: if the VLM judges a concept absent, its mask is suppressed to zero; otherwise, the original mask is preserved. This refinement yields clean and reliable annotations for training the concept-based modules. We provide more details of this process in App. C and App. D.

## 3.2 CONCEPT TOKENIZER

The Concept Tokenizer $\mathcal{T}_{concept}$ maps the hidden representations of the target model onto a constrained concept space. With the generated annotations, it is trained to predict two signals for each concept: a binary existence score and a spatial mask. As shown in the left part of Fig. 2, for each concept, the tokenizer first projects the internal feature maps into a dedicated latent space through a learnable transformation, producing concept-specific embeddings. Each embedding is aggregated to form holistic concept representations, which integrate evidence distributed over different channels or patches. Based on this unified representation, the tokenizer predicts two outputs: a binary existence score and a spatial mask localizing the concept within the input. We use two linear layers to compute the concept segmentation and the concept score separately. The formulation of this process is shown in Eq. 1 and Eq. 2.

$$s_i = \text{Sigmoid}\left(z_i^{(d_t)} \cdot W_{score,i}^{(d_t)} + b_{score,i}\right), 1 \le i \le n \tag{1}$$

$$m_i^{(d_s)} = z_i^{(d_t)} \cdot W_{seg,i}^{(d_t \times d_s)} + b_{seg,i}^{(d_s)}, 1 \le i \le n \tag{2}$$

where $s \in \mathbb{R}^n$ is the predicted concept score for the internal feature, and $m \in \mathbb{R}^{n \times d_s}$ is the predicted concept mask. The training loss is designed as the mean squared error (MSE) between the predicted concept score $s$, predicted concept mask $m$ and the true concept score $\mathcal{S}$, concept masks $\mathcal{M}$. We also apply an $L_1$ penalty to $W_{merge}$ to encourage sparsity, ensuring that each concept's representation is derived from a minimal set of channels or patches. The loss is formulated as Eq. 3.

$$\mathcal{L}_{tokenizer} = \lambda_1||\mathcal{S} - s||_2^2 + \lambda_2||\mathcal{M} - m||_2^2 + \lambda_3||W_{merge}||_1 \tag{3}$$

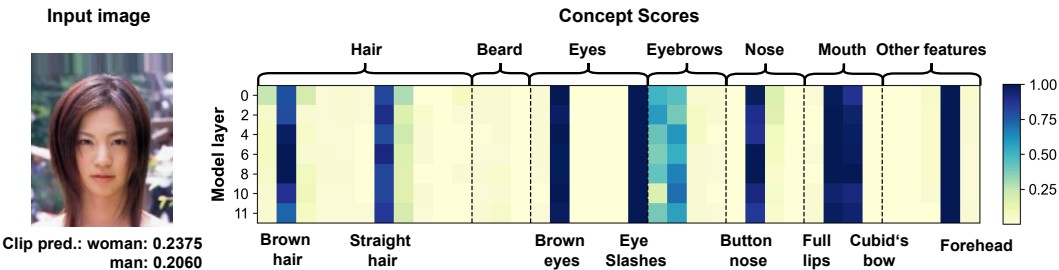

Figure 2: Computation process of concept tokenizer and concept aggregator.

Figure 3: Concept score example for an image. The y-axis is the layer of the model and x-axis is the concepts. The name of the concepts with high concepts score is shown at the bottom.

By doing so, $\mathcal{T}_{concept}$ enforces a direct alignment between intermediate features and human-interpretable concepts. An example of the predicted concept scores for different layers of a model is shown in Fig. 3.

### 3.3 CONCEPT AGGREGATOR

With the trained $\mathcal{T}_{concept}$, we freeze its parameters and further train a Concept Aggregator $\mathcal{A}_{concept}$, which is the decoder that reconstructs the original feature maps $h$ from the predicted concept representations. The right part of Fig. 2 shows its computation process in two steps. Firstly, it combines the information from the predicted concept score and segmentation by element-wise multiplication. The segmentation features of concepts absent from the image will be masked by low concept scores. Then, we utilize an MLP to fuse the predicted concept score $s$ and segmentation $m$ into a unified feature vector $f \in \mathbb{R}^{n \times d_m}$. Finally, we combines these concept features with a fully connected layer to produce the reconstructed feature map $\hat{h}_{concept}$. The training loss is designed as the MSE between the predicted feature $\hat{h}_{concept}$ and the truth feature $h$. Meanwhile, we want to encourage that every concept contributes to the channels or patches where it is retrieved. Therefore, we add a KL divergence loss between $W_{aggr}$ in $\mathcal{A}_{concept}$ and $W_{merge}$ in $\mathcal{T}_{concept}$, so that concept-channel distributions of the two parameters are aligned. We also add the $L_1$ loss to $W_{aggr}$ to encourage sparsity: each concept only contributes to the reconstruction of a few channels or patches. The loss function of $\mathcal{A}_{concept}$ is shown in Eq. 4, where $\text{sm}(\cdot)$ is the softmax function.

$$\mathcal{L}_{aggr} = \lambda_1 ||\hat{h}_{concept} - h||_2^2 + \lambda_2 \text{KL}(\text{sm}(W_{merge})||\text{sm}(W_{aggr}^\top)) + \lambda_3 ||W_{aggr}||_1 \tag{4}$$

### 3.4 FREE TOKENIZER & FREE AGGREGATOR

We introduce a Free Tokenizer and a Free Aggregator to discover features not covered by the predefined concept space. These two modules share the same architecture as the concept tokenizer–aggregator pair, but differ in training strategies: unlike the concept modules, they are trained jointly under the objective of the SAE with no external supervision. The training loss for the free tokenizer and the free aggregrator is designed to encode implicit concepts absent from the predefined concept pool and reconstruct the original features jointly with $\mathcal{T}_{concept}$ and $\mathcal{A}_{concept}$. The combination of the reconstructed features from the concept module and the free module is performed by a simple adding operation. Meanwhile, the $L_1$ loss is added to the output of the free tokenizer to preserve sparsity. The detailed loss is formulated in Eq. 5, with $\mathcal{T}_{free}$ and $\mathcal{A}_{free}$ as the free tokenizer

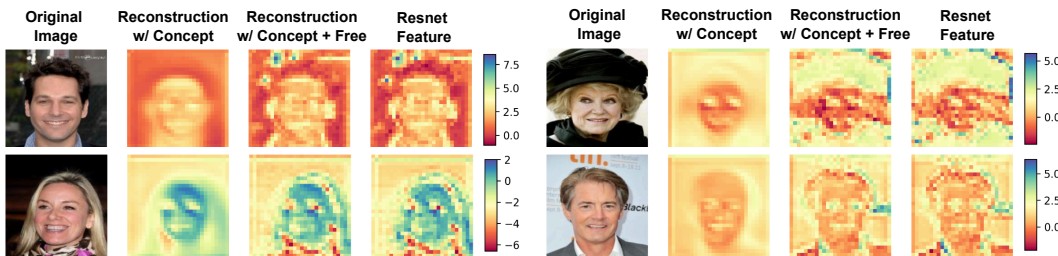

Figure 4: Reconstruction examples of our proposed method on ResNet features.

and the free aggregator separately. A comparison on reconstruction performance between the joint modules and the concept modules alone are shown in Fig. 4. We provide more visualization results of the reconstructed features at App. G. We provide some concept analysis of free tokens at App. F.

$$\mathcal{L}_{free} = \lambda_1 ||\mathcal{A}_{free}(\mathcal{T}_{free}(h)) + \hat{h}_{concept} - h||_2^2 + \lambda_2 ||\mathcal{T}_{free}(h)||_1 \tag{5}$$

## 4 EXPERIMENTS

To validate our proposed method, we conduct a series of experiments to investigate (1) whether the representations of concepts derived by **Concept-SAE** are faithful and distangled, and (2) **Concept-SAE**'s capabilities in model interpretation, error correction, and robustness analysis. Our evaluation is guided by the following research questions:

- **RQ1: Concept Focus & Faithfulness.** Do concept tokens faithfully capture model-internal, human-interpretable concepts—while remaining sparse and disentangled—compared to standard SAEs and concept-embedding baselines?
- **RQ2: Failure Diagnosis & Causal Correction.** Can Concept-SAE (1) identify concept patterns that differentiate correct vs. incorrect predictions, and (2) causally correct errors by intervening on concept scores/maps?
- **RQ3: Vulnerability Localization & Robustness Gains.** Can Concept-SAE accurately localize layers most susceptible to adversarial perturbations, and does targeting these layers for fine-tuning yield stronger robustness than standard choices?

### 4.1 EXPERIMENTAL SETUP

**Datasets.** We evaluate our approach on two datasets: **(1) CelebA** (Liu et al., 2015), we focus on the binary classification of the *Gender* attribute as the target label; and **(2) ImageNet-1k** (Deng et al., 2009), which involves classification across 1000 object categories.

**Models.** We consider two representative vision architectures: **(1) ResNet-18** (He et al., 2016), which is trained on CelebA and ImageNet-1k separately and subsequently analyzed; and **(2) Vision Transformer (ViT-B/32)** (Dosovitskiy et al., 2020; Radford et al., 2021), which is pre-trained on the LAION-2B dataset, and we evaluate the model with **Concept-SAE** in a zero-shot setting on both CelebA and ImageNet-1k.

### 4.2 EXPERIMENTS ON RESEARCH QUESTIONS

**RQ1: Concept Focus & Faithfulness.** We evaluate the purity of our learned concepts by comparing them with a concept-embedding baseline (CEM) (Espinosa Zarlenga et al., 2022), which represents each concept with two complementary embeddings supervised only by concept existence. For this experiment, we focus on concepts related to human appearance, with the full list provided in Table. 6 (Appendix). As shown in Fig. 5, our method reconstructs only the image regions directly associated with the target concepts. In contrast, the CEM baseline also reconstructs irrelevant background content (e.g., the text behind the person), indicating that its embeddings entangle unintended features. We further evaluate the localization ability of the concept tokens derived from our proposed

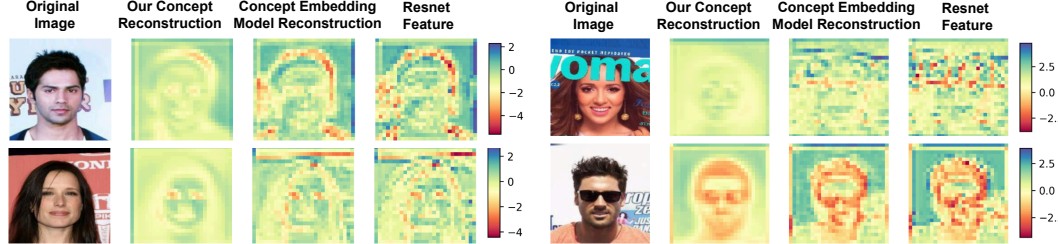

Figure 5: The difference between the vision model feature reconstruted by our concept module and concept embedding model.

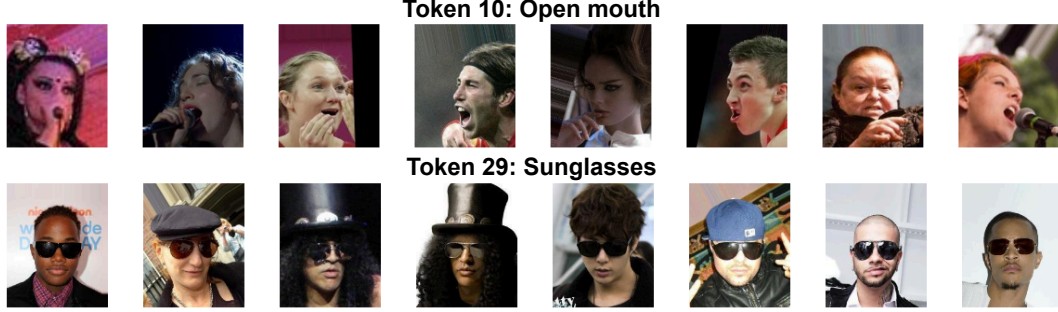

Figure 6: Two examples of free token visualizations, each with 8 most activated images from the CelebA dataset.

**Concept-SAE** model quantitatively. For the CelebA dataset, we construct two binary masks that separately cover the face region and the background. We then compute the reconstruction error of the vision model features within each region. To quantitatively measure localization, we define the **Localization Ratio** as the ratio between the MSE of the background and that of the face:

$$\text{LocR} = \frac{\text{MSE}\left((h - \hat{h}_{concept}) \odot M_{background}\right)}{\text{MSE}\left((h - \hat{h}_{concept}) \odot M_{face}\right)}. \tag{6}$$

where $M_{background}$ and $M_{face}$ are the binary masks for background and face part respectively. Since only facial concepts are used to train the SAE, a higher Localization Ratio shows the facial part is reconstructed better than the background part, which indicates a stronger ability to disentangle concepts and preserve their spatial localization. We compare our **Concept-SAE** with the CEM on shallow layers of ResNet, where localization information is still preserved. Deeper layers are excluded from evaluation, as their features are spatially fused and lack clear localization. As shown in Table 1, our **Concept-SAE** consistently achieves higher localization ratios than CEM, demonstrating superior concept localization. This highlights our method's superior ability to isolate and faithfully represent localized, semantically meaningful concepts. Moreover, introducing supervised concept tokens does not diminish the capacity of free tokens. As shown in Fig. 6, we can still get free tokens with rich semantic information through manual search (more examples in App. F). Therefore, in our proposed **Concept-SAE** framework, the free tokenizer and aggregator continue to capture residual, unconstrained features, enabling semantic analysis beyond predefined concepts.

**RQ2: Failure Diagnosis & Causal Correction.** By explicitly introducing concept scores, our method enables not only interpretability but also additional diagnostic capabilities. These scores can be used both to analyze model behavior and to directly adjust predictions. For **failure diagnosis**, we examine the difference patterns of concept scores for correctly predicted and mispredicted images. We interpret each concept score as the probability that the specific concept is represented internally by the model. To quantify reliability, we compute the entropy of concept scores across layers. Higher entropy indicates that the extracted feature of the vision model is ambiguous and less reliable, while lower entropy reflects more confident concept usage of the vision model. As shown in Table 2, for each layer of the vision model, incorrect predictions consistently exhibit higher entropy

Table 1: Localization ratio of our proposed **Concept-SAE** and CEM. Higher localization ratio indicates the concept derived has better localization characteristic and is better distangled.

| Model layer | LocR (Concept-SAE) | LocR (CEM) |
|---|---|---|
| ResNet-18 layer 5 | 1.472 | 1.019 |
| ResNet-18 layer 7 | 1.402 | 0.982 |
| ResNet-18 layer 9 | 1.395 | 1.002 |

Table 2: Information entropy of the concept score for different layers of the vision models. Higher information entropy indicates the output feature of that layer is more ambiguous. Red and blue numbers show the increase and decrease in entropy compared to the average of the original samples.

| Dataset | Model layer | Concept score entropy | | | |
|---|---|---|---|---|---|
| | | All pred. | Correct pred. | Incorrect pred. | Adversarial pred. |
| CelebA | ViT layer 0 | 0.249 | 0.248 | 0.273 (+0.024 ↑) | 0.320 (+ 0.071 ↑) |
| | ViT layer 2 | 0.234 | 0.234 | 0.257 (+ 0.023 ↑) | 0.353 (+ 0.119 ↑) |
| | ViT layer 4 | 0.223 | 0.222 | 0.246 (+ 0.023 ↑) | 0.379 (+ 0.156 ↑) |
| | ViT layer 6 | 0.216 | 0.216 | 0.235 (+ 0.019 ↑) | 0.346 (+ 0.130 ↑) |
| | ViT layer 8 | 0.197 | 0.197 | 0.212 (+ 0.015 ↑) | 0.291 (+ 0.094 ↑) |
| | ViT layer 10 | 0.197 | 0.196 | 0.203 (+ 0.007 ↑) | 0.256 (+ 0.059 ↑) |
| | ViT layer 11 | 0.208 | 0.207 | 0.221 (+ 0.013 ↑) | 0.284 (+ 0.076 ↑) |
| | ResNet-18 layer 5 | 0.289 | 0.287 | 0.320 (+ 0.031 ↑) | 0.298 (+ 0.009 ↑) |
| | ResNet-18 layer 7 | 0.284 | 0.281 | 0.313 (+ 0.029 ↑) | 0.294 (+ 0.010 ↑) |
| | ResNet-18 layer 9 | 0.279 | 0.277 | 0.307 (+ 0.028 ↑) | 0.315 (+ 0.036 ↑) |
| | ResNet-18 layer 12 | 0.277 | 0.274 | 0.304 (+ 0.027 ↑) | 0.305 (+ 0.028 ↑) |
| | ResNet-18 layer 14 | 0.274 | 0.271 | 0.302 (+ 0.028 ↑) | 0.324 (+ 0.050 ↑) |
| | ResNet-18 layer 17 | 0.253 | 0.249 | 0.289 (+ 0.036 ↑) | 0.304 (+ 0.051 ↑) |
| Imagenet | ViT layer 0 | 0.225 | 0.225 | 0.217 (- 0.008 ↓) | 0.171 (- 0.054 ↓) |
| | ViT layer 2 | 0.210 | 0.210 | 0.213 (+ 0.003 ↑) | 0.170 (- 0.040 ↓) |
| | ViT layer 4 | 0.198 | 0.198 | 0.207 (+ 0.009 ↑) | 0.213 (+ 0.015 ↑) |
| | ViT layer 6 | 0.180 | 0.180 | 0.206 (+ 0.026 ↑) | 0.187 (+ 0.007 ↑) |
| | ViT layer 8 | 0.163 | 0.163 | 0.184 (+ 0.021 ↑) | 0.174 (+ 0.011 ↑) |
| | ViT layer 10 | 0.158 | 0.158 | 0.160 (+ 0.002 ↑) | 0.165 (+ 0.007 ↑) |
| | ViT layer 11 | 0.166 | 0.166 | 0.186 (+ 0.020 ↑) | 0.192 (+ 0.026 ↑) |
| | ResNet-18 layer 5 | 0.224 | 0.224 | 0.226 (+ 0.002 ↑) | 0.235 (+ 0.011 ↑) |
| | ResNet-18 layer 7 | 0.215 | 0.214 | 0.219 (+ 0.004 ↑) | 0.227 (+ 0.012 ↑) |
| | ResNet-18 layer 9 | 0.185 | 0.185 | 0.191 (+ 0.006 ↑) | 0.186 (+ 0.001 ↑) |
| | ResNet-18 layer 12 | 0.190 | 0.188 | 0.196 (+ 0.006 ↑) | 0.192 (+ 0.002 ↑) |
| | ResNet-18 layer 14 | 0.188 | 0.188 | 0.195 (+ 0.007 ↑) | 0.203 (+ 0.015 ↑) |
| | ResNet-18 layer 17 | 0.169 | 0.169 | 0.175 (+ 0.006 ↑) | 0.182 (+ 0.013 ↑) |

in concept scores, suggesting that uncertain concept activations are a key factor of model failures. For **causal correction**, we intervene directly on the learned concept scores to modify the final output of the vision model. Specifically, we adjust selected concept scores to form a modified concept score vector, for instance, set the concept score of *beard* to zero. Then, the modified concept scores are passed through the aggregator with the original concept tokens to generate a counterfactual feature representation. We further substitute the original model feature with this generated counterfactual feature as the input so that we can modify the final output of the vision model. On CelebA misclassifications, for example, *male* images incorrectly predicted as *female* often exhibit insufficient activations of male-associated concepts such as beard, mustache, and adam's apple. By increasing the scores of these concepts, the prediction is corrected to male; conversely, reducing them corrects the opposite errors. As shown in Fig. 7, we find that interventions on deeper ViT layers are more effective, while in ResNet both shallow and deep layers yield strong corrections. These results demonstrate that **Concept-SAE** not only identifies ambiguous concept activations as the cause of

Table 3: Accuracy of the model on **adversarial samples** after finetuning. JS distance indicates the difference between the concept scores of the original samples and those of the adversarial samples in each layer. Red numbers show the increase in accuracy for adversarial samples after adversarial finetuning. Blue numbers show the decrease in JS distance for each layer of vision models before and after adversarial finetuning. Yellow background shows the top-3 values in that column.

| Dataset | Finetuned Model Layer | Adv Sample Accuracy | JS distance (before finetune) | JS distance (after finetune) |
|---|---|---|---|---|
| CelebA | None (ViT) | 70.05 % | - | - |
| | ViT layer 0 | 87.67% (+ 17.62% ↑) | 0.178 | 0.172 (- 0.006 ↓) |
| | ViT layer 2 | 80.03% (+ 9.98% ↑) | 0.137 | 0.103 (- 0.034 ↓) |
| | ViT layer 4 | 87.03% (+ 16.98% ↑) | 0.225 | 0.173 (- 0.052 ↓) |
| | ViT layer 6 | 83.06% (+ 13.01% ↑) | 0.160 | 0.125 (- 0.035 ↓) |
| | ViT layer 8 | 80.36% (+ 10.31% ↑) | 0.129 | 0.100 (- 0.029 ↓) |
| | ViT layer 10 | 76.88% (+ 6.83% ↑) | 0.121 | 0.064 (- 0.057 ↓) |
| | ViT layer 11 | 74.33% (+ 4.28% ↑) | 0.119 | 0.062 (- 0.057 ↓) |
| | All layers (ViT) | 93.32% (+ 23.27% ↑) | - | - |
| | None (ResNet) | 39.75% | - | - |
| | ResNet-18 layer 5 | 55.45% (+ 15.70% ↑) | 0.066 | 0.035 (- 0.031 ↓) |
| | ResNet-18 layer 7 | 61.19% (+ 21.44% ↑) | 0.080 | 0.049 (- 0.031 ↓) |
| | ResNet-18 layer 9 | 60.51% (+ 20.76% ↑) | 0.087 | 0.054 (- 0.033 ↓) |
| | ResNet-18 layer 12 | 63.53% (+ 23.78% ↑) | 0.095 | 0.072 (- 0.023 ↓) |
| | ResNet-18 layer 14 | 67.09% (+ 27.34% ↑) | 0.098 | 0.087 (- 0.012 ↓) |
| | ResNet-18 layer 17 | 68.08% (+ 28.33% ↑) | 0.109 | 0.106 (- 0.003 ↓) |
| | All layers (ResNet) | 80.52% (+ 40.77% ↑) | - | - |
| Imagenet | None (ViT) | 11.78 % | - | - |
| | ViT layer 0 | 28.24% (+ 16.46% ↑) | 0.070 | 0.052 (- 0.018 ↓) |
| | ViT layer 2 | 29.23% (+ 17.45% ↑) | 0.065 | 0.054 (- 0.011 ↓) |
| | ViT layer 4 | 29.78% (+ 18.00% ↑) | 0.067 | 0.043 (- 0.024 ↓) |
| | ViT layer 6 | 24.81% (+ 13.03% ↑) | 0.052 | 0.040 (- 0.012 ↓) |
| | ViT layer 8 | 25.77% (+ 13.99% ↑) | 0.051 | 0.041 (- 0.010 ↓) |
| | ViT layer 10 | 27.84% (+ 16.06% ↑) | 0.053 | 0.045 (- 0.008 ↓) |
| | ViT layer 11 | 25.24% (+ 13.46% ↑) | 0.048 | 0.040 (- 0.008 ↓) |
| | All layers (ViT) | 34.98% (+ 23.20% ↑) | - | - |
| | None (ResNet) | 9.51% | - | - |
| | ResNet-18 layer 5 | 13.87% (+ 4.36% ↑) | 0.030 | 0.017 (- 0.013 ↓) |
| | ResNet-18 layer 7 | 15.67% (+ 6.16% ↑) | 0.034 | 0.022 (- 0.012 ↓) |
| | ResNet-18 layer 9 | 17.20% (+ 7.69% ↑) | 0.039 | 0.027 (- 0.012 ↓) |
| | ResNet-18 layer 12 | 16.59% (+ 7.08% ↑) | 0.036 | 0.026 (- 0.010 ↓) |
| | ResNet-18 layer 14 | 14.89% (+ 5.38% ↑) | 0.038 | 0.027 (- 0.011 ↓) |
| | ResNet-18 layer 17 | 17.33% (+ 7.82% ↑) | 0.047 | 0.041 (- 0.006 ↓) |
| | All layers (ResNet) | 34.71% (+ 25.20% ↑) | - | - |

errors but also enables direct, causal interventions to repair predictions—capabilities not possible in prior SAE methods without explicit concept scores.

**RQ3: Vulnerability Localization & Robustness Gains.** We further leverage concept scores to systematically analyze how adversarial perturbations distort internal representations and to explore targeted defenses. Since adversarial attacks exploit weaknesses in feature representations, faithful concept scores should reveal both where and how these perturbations destabilize the model. To investigate this, we generate adversarial examples using FGSM (Goodfellow et al., 2014) and monitor changes in concept scores across the model hierarchy. An entropy-based analysis reveals two complementary effects: (1) some layers become excessively confident in a small set of concepts, as evidenced by reduced entropy, suggesting that the model latches onto spurious signals amplified

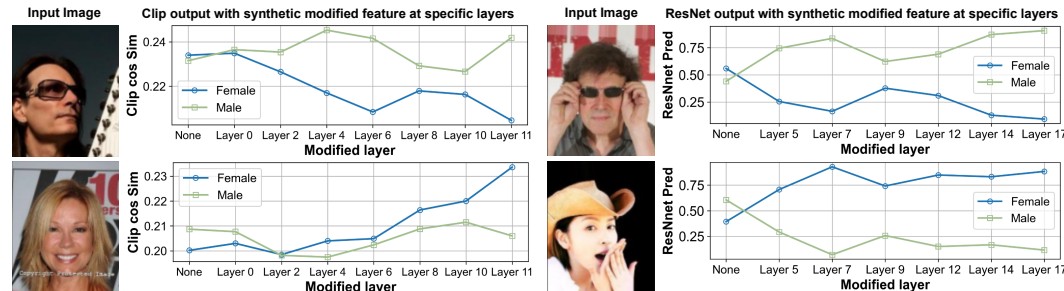

Figure 7: We modify the concept score and generate synthetic features through aggregator at different layers of the vision model. For male figures, we increase the concept score of *beard*, *adam's apple* to $1.0$. For female figures, we decrease the the concept score of *beard*, *adam's apple* to $0.0$. Then we examine how the vision models perform with these new features on failure images.

by the perturbation; (2) other layers display the opposite behavior, with increased entropy reflecting heightened uncertainty and confusion in concept extraction. These opposing tendencies, confidence collapse and semantic diffusion, jointly illustrate how adversarial perturbations erode the consistency of internal representations and disrupt the alignment between concepts and predictions. To more precisely quantify such changes, we compute the Jensen–Shannon (JS) distance (Lin, 2002) between the concept score distributions of clean and adversarial samples. Larger JS distances indicate stronger distributional shifts encountering adversarial samples and thus higher vulnerability. Based on this observation, we hypothesize that layers with higher JS distances are most fragile under attack. We validate this hypothesis through layer-wise finetuning. For each layer, we freeze the rest layers and retrain the chosen layer using mixed clean and adversarial samples for two epochs. As shown in Table 3, layers identified with larger JS distances consistently yield greater robustness improvements after finetuning compared to less vulnerable layers. These findings establish a direct link between our proposed vulnerability metric and effective defense strategies. Our analysis is both diagnostic and prescriptive: it not only identifies where adversarial perturbations compromise semantic integrity but also provides guidance for targeted interventions that significantly enhance robustness.

## 5 LIMITATIONS

Despite its effectiveness, **Concept-SAE** has several limitations. First, the accuracy of interpretation depends on the quality of the generated concept supervision. Noise or inaccuracies from the vision-language model (VLM) or the segmentation model may lead to imprecise concept identification and risk imposing misaligned interpretations onto the target model, though this issue is expected to diminish as foundation models continue to improve. Second, the reliance on spatial masks makes the framework particularly suited for concepts corresponding to localizable objects. Extending this approach to more abstract, textural, or globally distributed concepts—where spatial segmentation is inherently difficult—remains an important direction for future research.

## 6 CONCLUSION

Our work provides a blueprint for transforming mechanistic interpretability from a passive, correlational practice into an active, causal science. At the heart of this shift is **Concept-SAE**, whose hybrid disentanglement design produces high-fidelity, semantically validated concept tokens. These tokens enable rigorous experimentation: they establish direct causal links between internal concepts and predictions, diagnose failure modes by revealing unstable or ambiguous activations, and expose adversarial vulnerabilities that can be precisely targeted for intervention. By anchoring interpretability to faithful, localized representations, **Concept-SAE** elevates sparse autoencoders into reliable instruments for probing the inner causal mechanisms of neural networks. Looking forward, this framework represents more than a technical advance—it marks a foundational step toward a scientific paradigm in which neural networks are not merely observed but are actively probed, diagnosed, and debugged as systems governed by causal principles.

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

## A  USAGE OF LARGE LANGUAGE MODEL

We used Large language model (LLM) to aid and polish writing. We did not use LLM for other purposes.

## B  DETAILED TRAINING SETUP AND HYPERPARAMETERS

We adopt a three–stage training strategy for our proposed framework: **(1)** training the Concept Tokenizer, **(2)** training the Concept Aggregator, and **(3)** training the Free Tokenizer and Free Aggregator. All experiments are conducted on an Ubuntu 22.04 server equipped with an AMD EPYC 7K62 CPU and an NVIDIA A100-64G GPU. Below we describe each training stage in detail.

- **Concept Tokenizer Training.** The Concept Tokenizer is trained with dual supervision from concept existence scores and spatial masks. We use Adam as optimizer with an initial learning rate of $1 \times 10^{-3}$, scheduled by step learning rate scheduler with $\gamma = 0.1$ and step size 20, and train for 30 epochs. A batch size of 64 is applied. The loss function is shown below, combining existence score error, mask error, and an $L_1$ sparsity penalty, weighted by coefficients $(\lambda_1, \lambda_2, \lambda_3) = (1, 1, 0.1)$.

$$\mathcal{L}_{tokenizer} = \lambda_1 ||\mathcal{S} - s||_2^2 + \lambda_2 ||\mathcal{M} - m||_2^2 + \lambda_3 ||W_{merge}||_1 \qquad (7)$$

- **Concept Aggregator Training.** After freezing the tokenizer, we optimize the Concept Aggregator to reconstruct original features from predicted concept scores and masks. This stage uses Adam as optimizer with learning rate $1 \times 10^{-3}$, scheduled by Step learning rate scheduler with $\gamma = 0.1$ and step size 30, with batch size 64. It is trained for 50 epochs. The loss function is shown below. Our proposed loss combines feature reconstruction error, KL divergence aligning $W_{merge}$ and $W_{aggr}$, and an $L_1$ penalty on aggregator weights, weighted by coefficients $(\lambda_1, \lambda_2, \lambda_3) = (1, 0.01, 1)$.

$$\mathcal{L}_{aggr} = \lambda_1 ||\hat{h}_{concept} - h||_2^2 + \lambda_2 \mathrm{KL}(\mathrm{sm}(W_{merge})||\mathrm{sm}(W_{aggr}^\top)) + \lambda_3 ||W_{aggr}||_1 \quad (8)$$

- **Free Tokenizer and Free Aggregator Training.** Finally, the Free Tokenizer and Free Aggregator are trained jointly to capture residual features not represented by predefined concepts. We set the number of free tokens to be 36 for both the CelebA and the ImageNet dataset. We use Adam optimizer with a learning rate of $1 \times 10^{-3}$. It is trained for 30 epochs with batch size 64. The loss function is shown below. The loss objective combines reconstruction error with an $L_1$ sparsity penalty, weighted by $(\lambda_1, \lambda_2) = (1, 1)$.

$$\mathcal{L}_{free} = \lambda_1 ||\hat{h}_{free} + \hat{h}_{concept} - h||_2^2 + \lambda_2 ||\mathcal{T}_{free}(h)||_1 \qquad (9)$$

For clarity, Table 4 summarizes the hyperparameters across all training stages.

Table 4: Summary of hyperparameters for all training stages.

| Stage | Optimizer | Learning Rate | Epochs | Batch Size | Loss Weights |
|---|---|---|---|---|---|
| Concept Tokenizer | Adam | $1 \times 10^{-3}$ | 30 | 64 | $(1, 1, 0.1)$ |
| Concept Aggregator | Adam | $1 \times 10^{-3}$ | 50 | 64 | $(1, 0.01, 1)$ |
| Free Modules | Adam | $1 \times 10^{-3}$ | 30 | 64 | $(1, 1)$ |

## C  CONCEPT LABEL GENERATION DETAILS

In concept label generation, we query a large language model (LLM, GPT-4o (Achiam et al., 2023) in this paper) to generate a list of concepts and respective labels for each dataset. Specifically, we generate a list of attributes shared among multiple image categories in the dataset as concepts, and generate a list of tags for each attribute as the label set for the concept. We formulate detailed instructions to the LLM such that:

- The generated attributes are applicable to multiple categories in the dataset (though not necessarily related to all categories).
- The generated attributes are visually descriptive and can be clearly segmented within the image.
- The generated tags are descriptive, generalized, and not category-specific.

For the ImageNet dataset, we divide possible concepts into "animals", "plants", "man-made objects", and "background", and generate a concept set for each group. For the CelebA dataset with relatively simple settings, we generate the concept set directly for "human faces".

The instruction for ImageNet concept label generation is as follows:

*You are given the names of all image categories in the ImageNet dataset.*

*You need to generate a list of attributes for these categories.*

*- The generated attributes should be applicable to multiple categories. For example, "fur" is an appropriate attribute since it applies to multiple animal categories.*

*- The generated attributes should focus on physical regions and properties that are visually descriptive and can be clearly segmented within the image by image segmentation models like SAM (the Segment Anything Model). For example, attributes such as "color" and "texture" are too broad and subjective, making them less suitable for precise segmentation tasks. Instead, you should focus on specific, tangible features that can be distinctly identified and segmented in images.*

*You also need to generate a list of tags for each attribute.*

*- The generated tags should be descriptive and generalized. They should describe features like shapes, colors, textures, or structures that SAM can identify as distinct regions in the images, rather than class-specific terms.*

*Generate the attributes and tags for the "animal" categories / "plant" categories / "object" categories / "background" in the ImageNet dataset.*

*- The generated attributes and tags should be in the following JSON format:*

*{*

*    "attribute 1": ["tag11", "tag12", ...],*

*    "attribute 2": ["tag21", "tag22", ...],*

*    ...*

*}*

*- You should output the results in JSON format only, without any additional explanations or text.*

*- The total number of tags for all attributes should be around 50.*

The list of generated concept labels for ImageNet and CelebA are respectively shown in Table 5 and 6. The generated concepts provide highly comprehensive and fine-grained summarizations of core image characteristics in the datasets.

Table 5: The generated concept labels for the ImageNet dataset.

| Concepts | Concept Labels |
| --- | --- |
| *Animal* skin or fur | soft fur, coarse fur, short fur, long fur, fluffy fur, textured fur, striped fur, spotted fur, dotted fur, camouflage fur, zebra stripes fur, mottled fur, blotched fur, checkered fur, swirled fur, black animal, white animal, brown animal, gray animal, green animal, yellow animal, red animal, blue animal, orange animal, smooth skin, rough skin, slimy skin, scaly skin, wrinkled skin |
| *Animal* eyes | large eyes, small eyes, round eyes, oval eyes, wide-set eyes, bright eyes, piercing eyes, beady eyes, blue eyes, green eyes, brown eyes, red eyes, black eyes, white eyes |
| *Animal* mouth | sharp teeth, flat mouth, wide mouth, small mouth, open mouth, pointed mouth, sharp teeth, pointed teeth, flat teeth, large teeth, small teeth, sharp beak, curved beak, pointed beak, wide beak, red mouth, pink mouth, black mouth, white mouth, brown mouth |
| *Animal* nose | pointed nose, long nose, round nose, flat nose, snout, nostrils |
| *Animal* ears | pointed ears, large ears, small ears, floppy ears, curved ears, round ears |
| *Animal* limbs | muscular legs, long legs, short legs, slender legs, four legs, bipedal, quadrupedal, jointed legs, thick limbs, slender limbs, feathered wings, bat wings, large wings, small wings, flapping wings, folded wings, open wings |
| *Animal* tail | long tail, short tail, fluffy tail, curved tail, pointed tail, furry tail, lashing tail, thick tail, thin tail |
| *Animal* claws and feet | padded paws, webbed feet, hooved feet, clawed feet, sharp claws, large paws, small paws, sharp claws, curved claws, long claws, short claws |
| *Plant* leaves | green leaves, broad leaves, narrow leaves, small leaves, large leaves, serrated edges, smooth edges, pointed leaves, oval leaves, heart-shaped leaves, round leaves, lance-shaped leaves, needle-like leaves, palm-shaped leaves |
| *Plant* flowers | bright flowers, petaled flowers, yellow flowers, red flowers, white flowers, purple flowers, pink flowers, blue flowers, large flowers, small flowers, orange flowers, flower buds, open flowers, closed flowers |
| *Plant* branches | branching, straight branches, curved branches, dense branches, sparse branches, long branches, short branches |
| *Plant* stems and trunks | woody stem, green stem, branched trunk, straight trunk, thick trunk, thin trunk |
| *Man-made objects* | round object, oval object, rectangular object, square object, elongated object, curved object, triangular object, angular object, smooth surface, rough texture, polished surface, matte finish, glossy surface, shiny texture, coarse surface, grainy texture, slippery surface, bumpy texture, wood material, metal material, plastic material, stone material, fabric material, rubber material, glass material, ceramic material, paper material, leather material, striped object pattern, dotted object pattern, checkered object pattern, plaid object pattern, swirled object pattern, zigzag object pattern, camouflage object pattern, polka-dotted object pattern, geometric object pattern, floral object pattern, red object, blue object, green object, yellow object, orange object, purple object, pink object, black object, white object, gray object, brown object |
| *Background* | blurred background, sky background, mountain background, forest background, river background, lake background, sea background, urban background, beach background, desert background, snow background, greenery background, indoor background |

Table 6: The generated concept labels for the CelebA dataset.

| Concepts | Concept Labels |
|---|---|
| Hair | blond hair, brown hair, black hair, red hair, gray hair, white hair, straight hair, wavy hair, curly hair, coily hair, receding hairline |
| Beard | beard, mustache, goatee |
| Eyes | blue eyes, brown eyes, green eyes, hazel eyes, gray eyes, eyelashes |
| Eyebrows | arched eyebrows, straight eyebrows, thick eyebrows, thin eyebrows |
| Nose | roman nose, button nose, aquiline nose, upturned nose |
| Mouth | thin lips, full lips, cupid's bow, dimples |
| Freckles | freckles |
| Scars | scars |
| Adam's apple | adam's apple |
| Forehead | forehead |
| Wrinkles | wrinkles |

# D    CONCEPT SEGMENTATION GENERATION DETAILS

We leverage ClipSeg (Lüddecke & Ecker, 2022) as the image segmentation model. We use image and concept label as input, then it will output the segmentation of the concept label on the image. As shown in Fig. 8 and Fig. 9, we provided some segmentation results provided by ClipSeg.

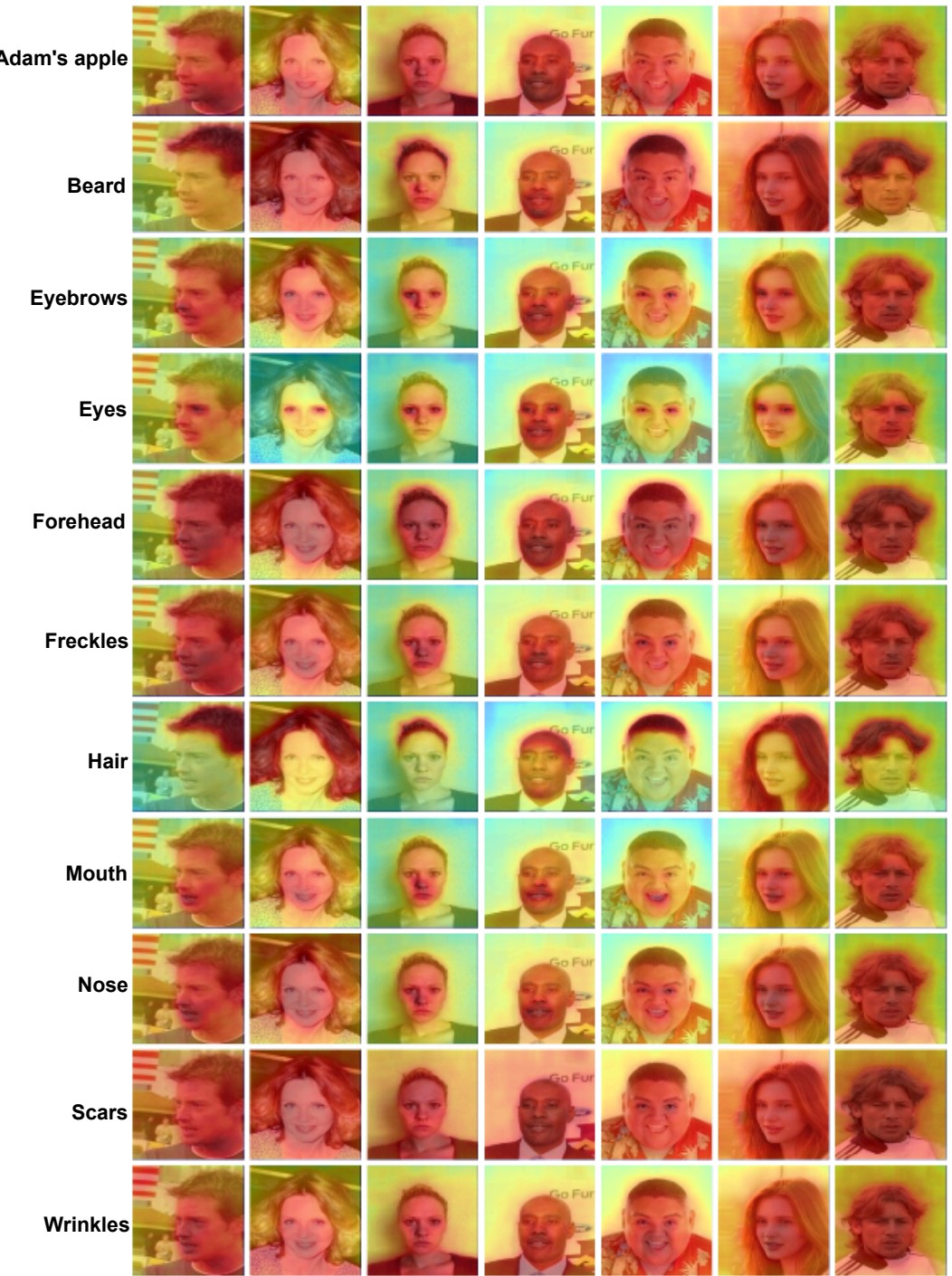

Figure 8: Examples of segmentation results provided by ClipSeg from the CelebA dataset.

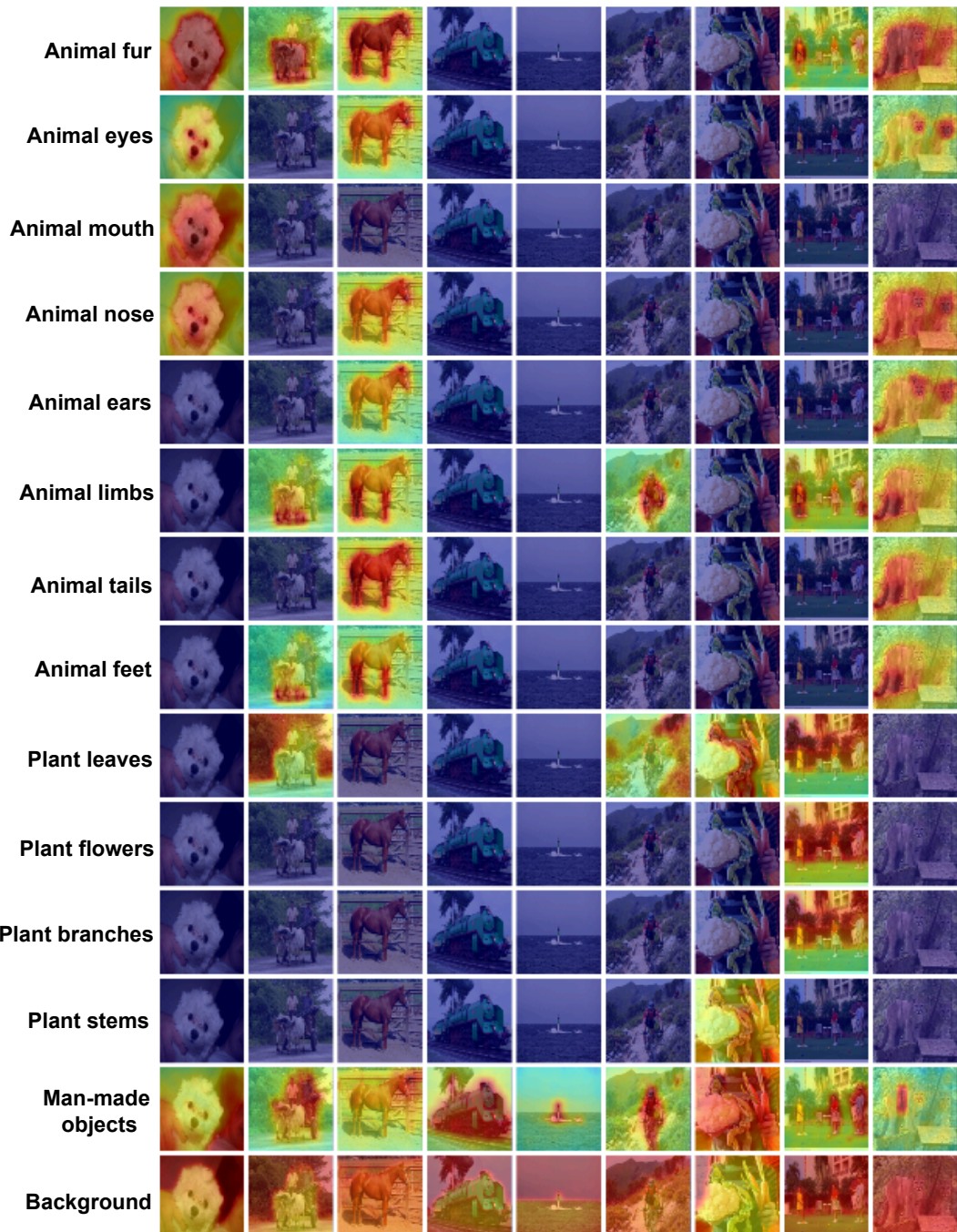

Figure 9: Examples of segmentation results provided by ClipSeg from the ImageNet dataset.

# E  INSPECTING THE INFLUENCE OF IRRELEVANT CONCEPTS

We evaluate our model using a set of irrelevant concepts and analyze the results on both datasets. Here, irrelevant concepts refer to those whose concept scores $s$ remain nearly zero across most images in the dataset. To assess their effect, we compute the entropy of the concept scores produced by our model at each layer. As shown in Table. 7, the entropy values for irrelevant concepts are consistently close to zero, indicating that these concepts provide no meaningful information about the model's internal computation. This finding suggests that incorporating irrelevant concepts leads to non-informative, non-analyzable results and should therefore be avoided in practice.

Table 7: Information entropy of the concept score for different layers of the vision models calculated with relevant and irrelevant concepts.

| Dataset | Model layer | Relevant concept score entropy | Irrelevant concept score entropy |
|---|---|---|---|
| CelebA | ViT layer 0 | 0.249 | $9.95 \times 10^{-5}$ |
| | ViT layer 2 | 0.234 | $3.75 \times 10^{-5}$ |
| | ViT layer 4 | 0.223 | $9.16 \times 10^{-5}$ |
| | ViT layer 6 | 0.216 | $2.67 \times 10^{-4}$ |
| | ViT layer 8 | 0.197 | $9.15 \times 10^{-5}$ |
| | ViT layer 10 | 0.197 | $5.87 \times 10^{-6}$ |
| | ViT layer 11 | 0.208 | $4.35 \times 10^{-6}$ |
| | ResNet-18 layer 5 | 0.289 | $9.35 \times 10^{-5}$ |
| | ResNet-18 layer 7 | 0.284 | $1.61 \times 10^{-4}$ |
| | ResNet-18 layer 9 | 0.279 | $2.11 \times 10^{-4}$ |
| | ResNet-18 layer 12 | 0.277 | $1.19 \times 10^{-4}$ |
| | ResNet-18 layer 14 | 0.274 | $6.53 \times 10^{-4}$ |
| | ResNet-18 layer 17 | 0.253 | $2.63 \times 10^{-5}$ |
| Imagenet | ViT layer 0 | 0.225 | $1.37 \times 10^{-4}$ |
| | ViT layer 2 | 0.210 | $5.93 \times 10^{-5}$ |
| | ViT layer 4 | 0.198 | $9.63 \times 10^{-5}$ |
| | ViT layer 6 | 0.180 | $1.56 \times 10^{-5}$ |
| | ViT layer 8 | 0.163 | $3.23 \times 10^{-5}$ |
| | ViT layer 10 | 0.158 | $9.31 \times 10^{-5}$ |
| | ViT layer 11 | 0.166 | $8.24 \times 10^{-4}$ |
| | ResNet-18 layer 5 | 0.224 | $1.72 \times 10^{-4}$ |
| | ResNet-18 layer 7 | 0.215 | $6.19 \times 10^{-5}$ |
| | ResNet-18 layer 9 | 0.185 | $3.71 \times 10^{-5}$ |
| | ResNet-18 layer 12 | 0.190 | $6.54 \times 10^{-5}$ |
| | ResNet-18 layer 14 | 0.188 | $3.89 \times 10^{-4}$ |
| | ResNet-18 layer 17 | 0.169 | $3.45 \times 10^{-4}$ |

## F  CONCEPT OF FREE TOKENS

We mannually find some free tokens trained in our framework that have visible semantic concepts. For each free token, we select the 16 most activated images from the dataset for visualization. The visulization is shown in Fig. 10 and Fig. 11.

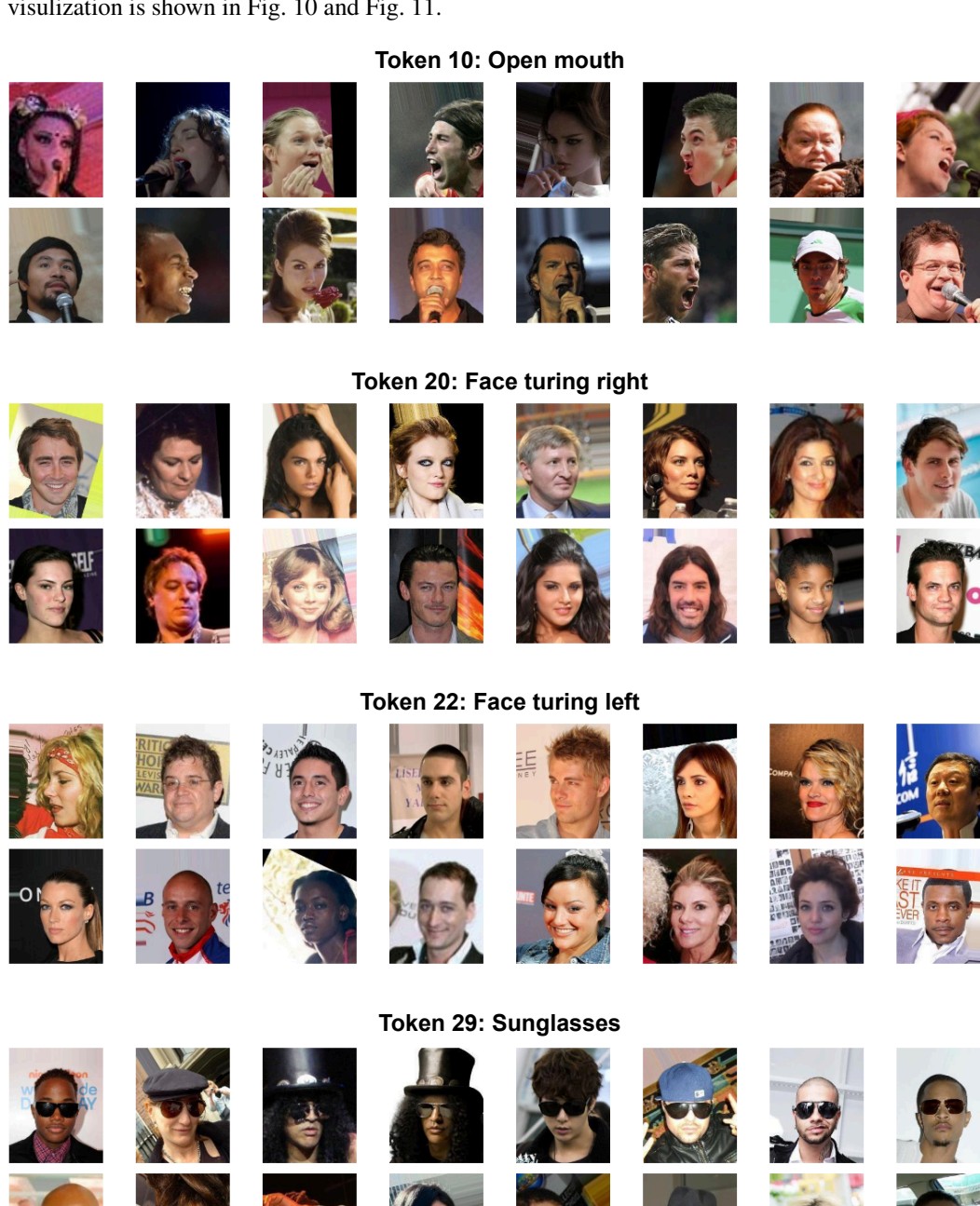

Figure 10: More examples of free token visualizations, each with 16 most activated images from the CelebA dataset.

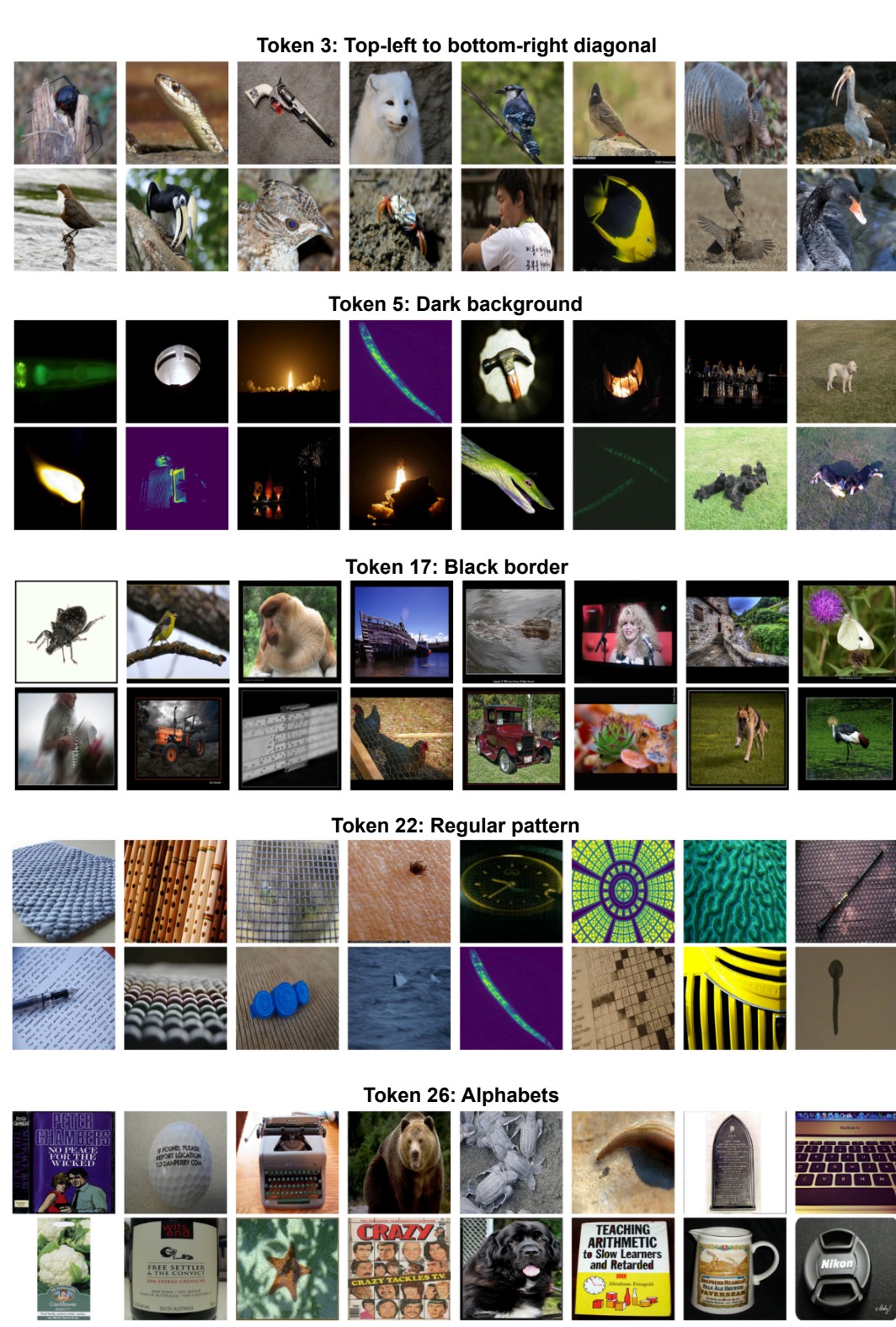

Figure 11: More examples of free token visualizations, each with 16 most activated images from the ImageNet dataset.

# G EXTENDED RESULTS OF FEATURE RECONSTRUCTION

As shown in Fig. 12 and Fig. 13, we provide some more reconstruction examples using our proposed model.

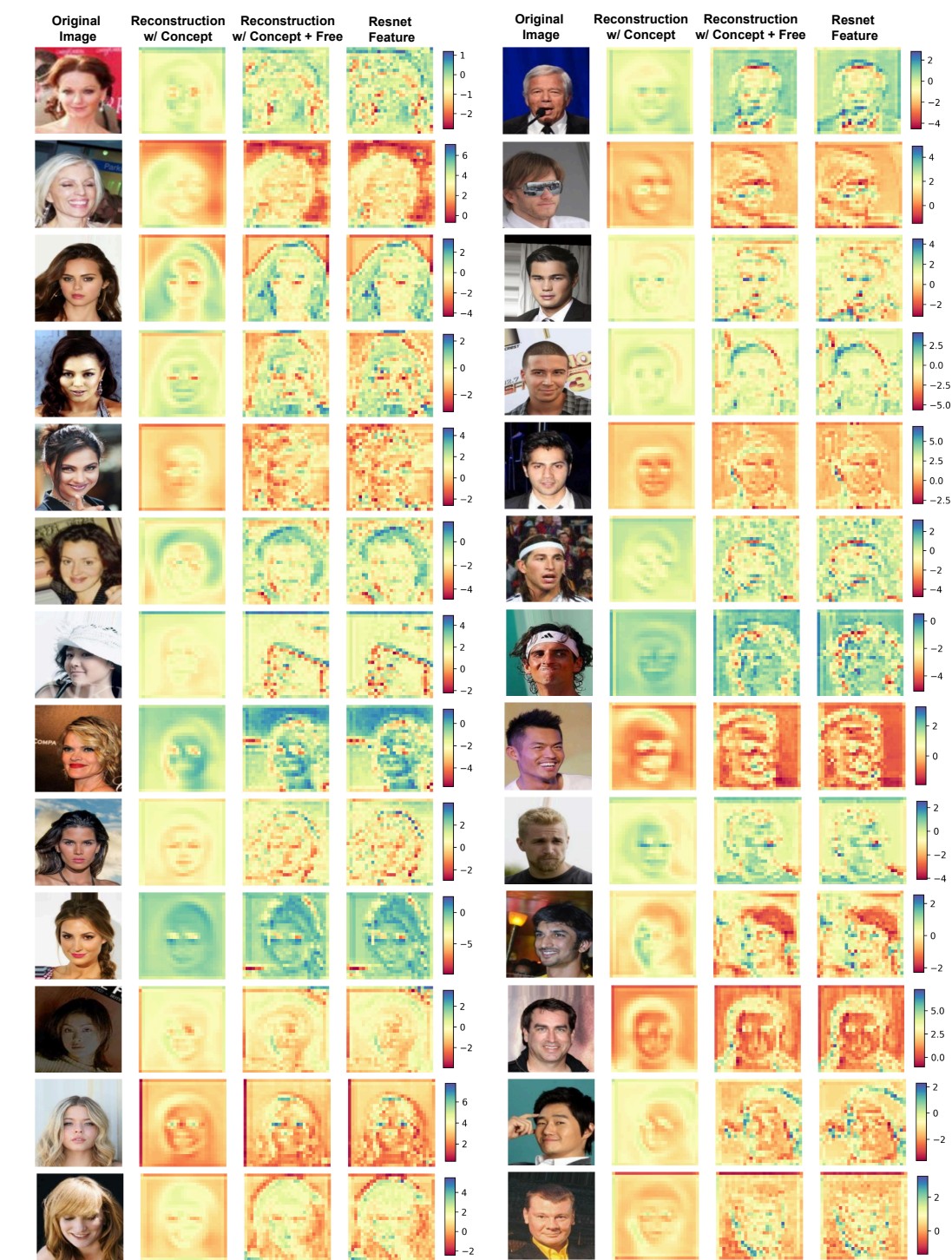

Figure 12: Reconstruction examples for the CelebA dataset.

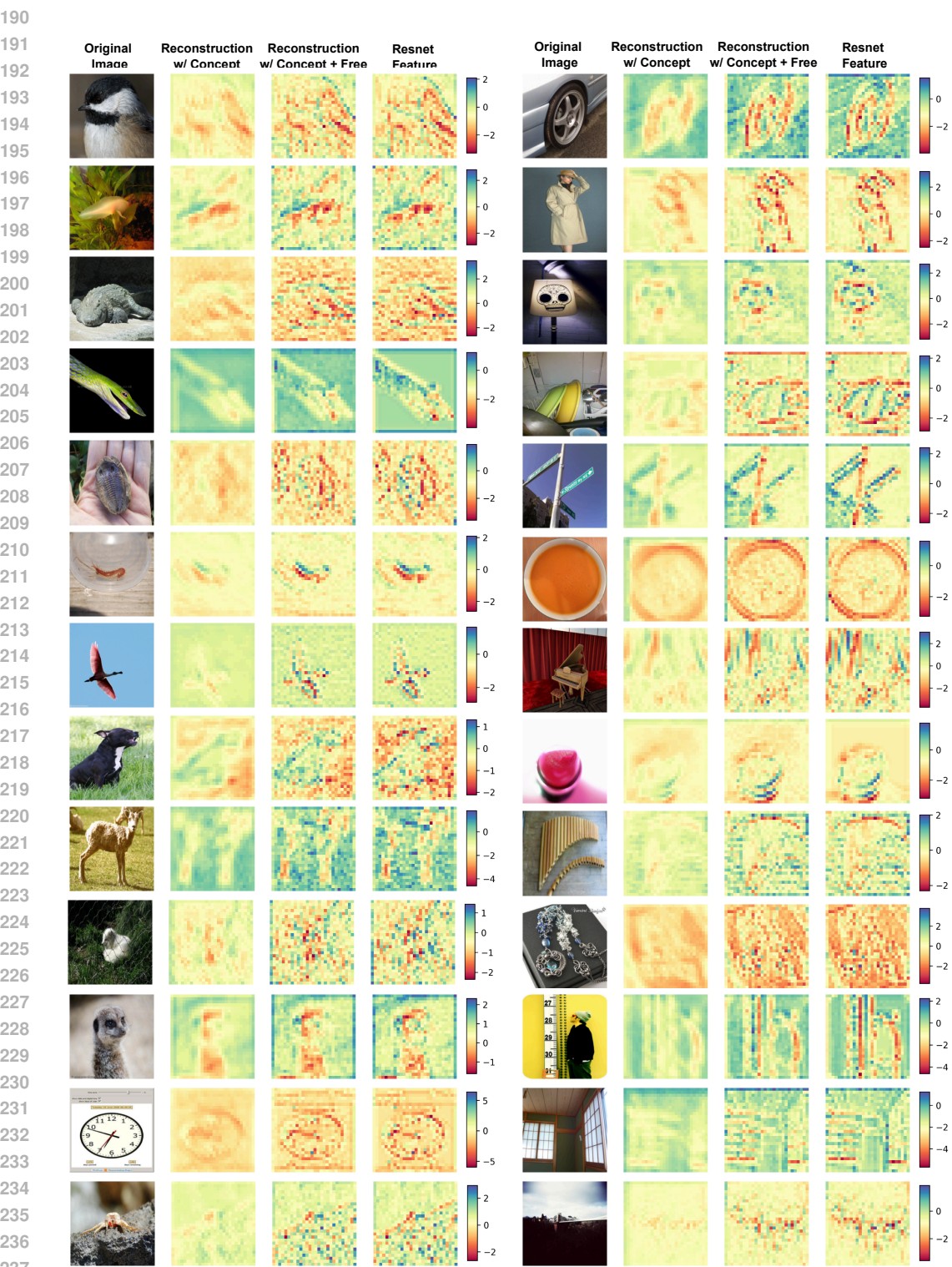

Figure 13: Reconstruction examples for the ImageNet dataset.

