# OpenReview forum: "Concept-SAE: Active Causal Probing of Visual Model Behavior"
_ICLR.cc/2026/Conference — ICLR 2026 Conference Withdrawn Submission_

### Official Review · Reviewer_fBz6 · 2025-10-29

**Soundness:** 2
**Presentation:** 2
**Contribution:** 2
**Rating:** 4
**Confidence:** 3

**Summary:**

Concept-SAE presents a novel and well-structured framework for moving Sparse Autoencoders (SAEs) from correlational interpretation toward causal probing, the work still suffers from several conceptual and methodological weaknesses. These issues limit the strength of its causal claims and the general reliability of its conclusions.
Concept-SAE introduces a novel framework that transforms sparse autoencoders from passive interpretability tools into active instruments for causal probing by grounding learned features in human-defined, spatially localized concepts.
The method is innovative and well-executed, demonstrating strong interpretability and diagnostic capabilities, but its causal claims rely heavily on external supervision and lack formal theoretical grounding.

**Strengths:**

1. Concept-SAE extends conventional SAEs with a hybrid disentanglement mechanism that combines: 1)supervised concept tokens (grounded in human-defined semantics) and 2) unsupervised free tokens (for residual discovery).

**Weaknesses:**

1.Concept supervision in Concept-SAE depends entirely on external models such as CLIP and CLIPSeg. Any bias or error in these models propagates directly into the learned “concept tokens,” which means the method effectively explains the inductive biases of CLIP, not the true internal structure of the target network. It becomes a second-order interpretation mediated by external semantic priors.

2. Although the authors retain “free tokens,” the dual supervision on existence and spatial masks strongly constrains the latent space.
No quantitative evidence is provided that concept tokens and free tokens remain disentangled or independent.

3. The predefined concept list is generated automatically using GPT-4o, without human validation or quality control.
This process can produce overlapping, redundant, or poorly defined concepts.

4. All experiments are conducted on ResNet-18 and ViT-B/32 within visual tasks (CelebA and ImageNet-1k).
No experiments test generalization to non-visual or multimodal models.

5. The causal intervention results (e.g., toggling “beard” or “hair” scores) are shown qualitatively, without repeated trials, statistical tests, or confidence intervals. The relationship between intervention magnitude and prediction change is not formally analyzed.

6. Although the paper repeatedly refers to “causal probing” and “counterfactual features,” no formal definition (e.g., structural causal model, do-calculus) is provided. The interventions operate in feature space without causal grounding.

**Questions:**

q1: Layer-wise interventions modify internal features at a single level, but the authors do not analyze how these changes propagate to other layers.

q2: The predefined concept list is generated automatically using GPT-4o, without human validation or quality control.
This process can produce overlapping, redundant, or poorly defined concepts.

q3: The work’s causal narrative remains metaphorical rather than formalized; Concept-SAE performs feature manipulation, not true causal inference.

---

### Official Review · Reviewer_63nn · 2025-10-31

**Soundness:** 1
**Presentation:** 2
**Contribution:** 1
**Rating:** 2
**Confidence:** 3

**Summary:**

The paper introduces Concept-SAE as an alternative to regular Sparse Autoencoders. Concpet-SAE includes a supervised component using a VLM and a segmentation model as ground truth, as well as an unsupervised "free Autoencoder" component akin to an SAE. The authors introduce this novel architecture and analyse the interpretability of both components. They also include results with interventions on the concept scores to change the model predictions and the effect of adversarial examples on the entropy of concept scores.

**Strengths:**

- The paper introduces a novel architecture with several interesting components.
- The authors evaluate several aspect of this method across 2 different models (Resnet and ViT)

**Weaknesses:**

- **Lack of clarity:**  Some concepts or parts of the method are not properly defined, for example what W_proj and W_merge are needs to be deduced from context and the dimension indications in Figure 2. Lack of descriptive captions in information dense figures like Figure 1 or Figure 2 makes the paper harder to read, particularly since some of the details are not clearly explained in the main text. For example, it is not clear how the different heatmaps are obtained, do they show the position at which the feature fires in the convolutional case or some kind of feature attribution? Additionally, if Concept-SAE is a kind of SAE, it would be helpful to explain it at some point in terms of encoder and decoder. I personally find “Concept tokenizer” to be a confusing name for the first half of the architecture, particularly since, if I understand correctly, it merges representations across different tokens/patches in the ViT case. Finally, the takeaways from figures 4 and 5 are unclear. What is the “Resnet Feature”? Are the activations/attributions meant to be positive for target concepts like eyes and hair? This does not seem to be the case for bottom left and top right concept reconstructions in Figure 5.

- **Lack of a baseline for comparison:**  There is no comparison to obvious baselines like linear probes or regular SAEs. No clear argument is made for why this should be used instead of regular SAEs beyond the causality claims which I find unconvincing as outlined below.

- **Causality claims:** The author claim Concept-SAE enable causal interventions to repair predictions “capabilities not possible in prior SAE methods”. However, it is not clear how this is different from the common technique of using SAE features or linear probes to steer models, which could also be used to artificially activate the beard concept leading to classification as male. In light of this, claims like “Our work provides a blueprint for transforming mechanistic interpretability from a passive, correlational practice into an active, causal science.” feel overstated.

**Questions:**

The method has many components which are interesting but feel a bit ad-hoc, maybe this could be alleviated by a clearer motivation of each component. Particularly, what motivates the departure from a more straight-forward approach like training supervised linear probes for the target concepts and a normal SAE for the rest (akin to your free autoencoder)? Presumably, a linear probe trained to detect eyes would activate or be attributed to the parts of an image containing eyes, what is the advantage of training the concept segmentation in parallel?

---

### Official Review · Reviewer_4r4b · 2025-11-01

**Soundness:** 3
**Presentation:** 2
**Contribution:** 2
**Rating:** 2
**Confidence:** 4

**Summary:**

This paper introduces Concept-SAE, a sparse autoencoder (SAE) framework that injects “concept tokens” into the SAE latent space. These concept tokens are designed to be more semantically grounded and disentangled than traditional SAE features, which are not well suited for causal probing of model behavior. The method and experiments are focused on the vision domain. The framework uses a "dual supervision" strategy, leveraging an external VLM for concept existence scores and a segmentation model for concept spatial masks. This is contrasted with "free tokens" that capture residual, unsupervised features. Concept-SAE is able to localize adversarial vulnerabilities to specific layers.

**Verdict**: While the problem space is important, the proposed method does not appear to offer a significant advantage over existing methods, as it relies on predefined, segmentable concepts rather than novel feature discovery. The paper's claims of unique causal capabilities are not convincingly supported by the experimental results.

**Strengths:**

- S1: The problem space of identifying truly meaningful and grounded features in deep learning model internals is indeed important. It is true that current SAE implementations yield features that are essentially observational in nature.
- S2: The Concept-SAE approach does offer advantages over post-hoc analysis of SAE features and activation maps.

**Weaknesses:**

Major:
- W1: The proposed method seems to very narrowly define “concept feature”, by restricting "concepts" to only those that are (1) predefined by an external VLM and (2) spatially segmentable. This excludes a vast range of more abstract features that mechanistic interpretability aims to find, somewhat defeating the purpose of using an SAE in the first place. If you already pre-define concepts that you care about, why use an SAE at all? Just use a plain classifier or segmentation model to find these “features”.
- W2: Similar to W1, there don’t appear to be synergies between the “concept tokens” and the “free tokens” in the implementation of the method. For instance, for the “beard concept”, is this present in both the concept tokens AND the free tokens? If so, how would this concept differ between the two? Should we analyze the “beard concept token” or the “beard free token”? The authors state “These results demonstrate that Concept-SAE not only identifies ambiguous concept activations as the cause of errors but also enables direct, causal interventions to repair predictions—capabilities not possible in prior SAE methods without explicit concept scores.” Why should this be the case? There is a large amount of activation steering literature using SAE concepts that shows that you can use prior SAE concepts to perform causal interventions. If I am mistaken, the authors should elaborate more.
- W3: The results presented are not entirely convincing. For instance, the authors state “As shown in Fig. 5, our method reconstructs only the image regions directly associated with the target concepts”. This is not exactly obvious just from looking at the images in Fig.5, and anyway this would be a fairly obvious result since the concept features are derived explicitly from segmentation masks. Further, in Table 2, are the numbers presented in Table 2 average entropy across a dataset? If so, standard deviations should be provided, which may overlap with the very modest increase in accuracy between the correct and incorrect predictions. RQ3 presents the most compelling findings that these concept features are useful, but again the results presented are somewhat simplistic. It is fairly obvious that turning up “beard” concepts on male images would lead to a higher confidence in the model predicting “male”. A more interesting test would be composing different concepts, swapping concepts, and testing robustness in that way.

Minor:
- W4: The Figure 1 caption needs to be expanded. It is not clear what each of the symbols correspond to, or how each step relates to the other.
- W5: The generic SAE formulation and objective function should be explained in the Methodology section. This would help with readability.

**Questions:**

- Q1: The "concept tokens" are heavily supervised using predefined, spatially-segmentable concepts. Given this, what is the specific advantage of using a sparse autoencoder framework for these tokens at all, as opposed to simply training a direct concept classifier or segmentation model on the target model's internal activations?
- Q2: The paper claims prior SAE methods are "not possible" to use for causal interventions. Could the authors elaborate on this, especially given existing work on activation steering using standard SAE features? What specific property of "explicit concept scores" enables interventions that standard, well-disentangled SAEs supposedly lack?
- Q3: What is the relationship between the supervised concept tokens and the unsupervised free tokens? For example, if a "beard" concept is captured by a concept token, could a similar feature also be learned by a free token? If this redundancy exists, which token should be used for a causal intervention, and how do their roles differ?
- Q4: In Table 2, you show that concept score entropy is higher for incorrect predictions versus correct ones. Are these reported values averages across the test set? If so, could you provide the standard deviations to demonstrate that these differences are statistically significant?

---

### Official Review · Reviewer_yaMg · 2025-11-02

**Soundness:** 3
**Presentation:** 3
**Contribution:** 2
**Rating:** 4
**Confidence:** 3

**Summary:**

The paper proposes a novel method to explain the ResNet-18/ViT model's understanding of images. A vision language model is used to generate concepts, and they are then merged with the output of an image segmentation model. The proposed Concept-SAE is then trained on the ResNet/ViT model activation to obtain a mapping of the learned concepts.

**Strengths:**

1. The idea of using LLM to create concepts and then merging them with image segmentation is novel and needs further study.
2. The idea of probing the vision model was not trained on, for example, "fur" is quite interesting.

**Weaknesses:**

1. Concept-SAE model quantitatively only talks about finding faces and not the concepts in it during reconstruction. It is not clear if the localization ratio is the right way to measure success.
2. It is not clear why concepts are needed for ResNet and related architectures, if the models themselves are not trained for those concepts.
3. For the third research question, "Vulnerability Localization & Robustness Gains". The proposed method provides some insight but does not outperform other methods in improving robustness. There is no empirical evidence provided here.
4. Overall, the issue is that for all three research questions, there is no empirical comparison with the SOTA showing the method outperforms them.

**Questions:**

1. Is there a way to compare the results of the paper with existing work and show that it outperforms them on some metric? Overall, I like the idea, but I need some more empirical results to accept the paper.

---

### Note · Authors · 2025-11-12

I have read and agree with the venue's withdrawal policy on behalf of myself and my co-authors.